# Predictor of Early Administration of Antibiotics and a Volume Resuscitation for Young Infants with Septic Shock

**DOI:** 10.3390/antibiotics10111414

**Published:** 2021-11-19

**Authors:** Osamu Nomura, Takateru Ihara, Yoshihiko Morikawa, Hiroshi Sakakibara, Yuho Horikoshi, Nobuaki Inoue

**Affiliations:** 1Department of Emergency and Disaster Medicine, Hirosaki University, Hirosaki 036-8562, Japan; 2Division of Pediatric Emergency Medicine, Tokyo Metropolitan Children’s Medical Center, Tokyo 183-8561, Japan; ihara.er@gmail.com; 3Clinical Research Support Center, Tokyo Metropolitan Children’s Medical Center, Tokyo 183-8561, Japan; yoshihiko_morikawa@tmhp.jp; 4Division of General Pediatrics, Department of Pediatrics, Tokyo Metropolitan Children’s Medical Center, Tokyo 183-8561, Japan; hiroshi_sakakibara@tmhp.jp; 5Division of Infectious Diseases, Department of Pediatrics, Tokyo Metropolitan Children’s Medical Center, Tokyo 183-8561, Japan; yuho74@hotmail.com; 6Department of Human Resources and Development, National Center for Global Health and Medicine, Tokyo 162-8655, Japan; nobuinoue@me.com

**Keywords:** sepsis, infants, 1-h bundle, tachycardia

## Abstract

(1) Background: It is critical to administer antibiotics and fluid bolus within 1 h of recognizing sepsis in pediatric patients. This study aimed to identify the predictor of the successful completion of a 1-h sepsis bundle for infants with suspected sepsis. (2) Methods: This is an observational study using a prospective registry including febrile young infants (aged < 90 days) who visited a pediatric emergency department with a core body temperature of 38.0 °C or higher and 36.0 °C or lower. Univariate and logistic regression analyses were conducted to determine the predictor (s) of successful sepsis bundle completion. (3) Results: Of the 323 registered patients, 118 patients with suspected sepsis were analyzed, and 38 patients (32.2%) received a bundle-compliant treatment. Among potential variables, such as age, sex, and vital sign parameters, the logistic regression analysis showed that heart rate (odds ratio: OR 1.02; 95% confidence interval: 1.00–1.04) is a significant predictor of the completion of a 1-h sepsis bundle. (4) Conclusions: We found that tachycardia facilitated the sepsis recognition and promoted the successful completion of a 1-h sepsis bundle for young infants with suspected septic shock and a possible indicator for improving the quality of the team-based sepsis management.

## 1. Introduction

Young infants (aged < 90 days) have a high risk of developing serious infections leading to sepsis due to their greater susceptibility to pathogens. In the past four decades, enormous efforts have been made to develop an evidence-based approach for evaluating young febrile infants [1]. Some research groups have aimed to develop prediction models including clinical variables, such as age, and laboratory examinations, including peripheral white blood cell counts (WBCs) and urine tests, to rule out serious bacterial infections (SBIs) [2,3]. Other groups have proposed low-risk criteria based on the assumption that ruling out SBIs was more reasonable by considering resource utilization, as the clinical performance of the developed prediction models was low [4]. Recent studies, however, have developed new prediction models using machine learning technologies and new biomarkers, such as procalcitonin, and show excellent performance compared to previous models [5,6,7]. Nevertheless, most of the proposed criteria and prediction models focus on managing well-appearing infants and do not refer to the treatment of sepsis infants. It is also important to uncover evidence to improve emergency care for critically ill infants with sepsis.

The Surviving Sepsis Campaign International Guidelines for the Management of Septic Shock and Sepsis-associated Organ Dysfunction in Children recommend the early recognition of sepsis and completion of a “bundle of therapy” for sepsis treatment in children. The bundle includes sampling a blood culture and the administration of antibiotics and fluid bolus within 1 h of septic shock recognition [8]. 

Recent studies have shown that bundle-compliant care in septic shock children was associated with lower mortality and that the delayed administration of antibiotics was conversely associated with increased mortality rates [9,10]. In this context, the guidelines recommend implementing a standardized protocol for managing septic shock among children in each institution [8].

The completion of the 1-h sepsis bundle may be more challenging in young infants than it is in older children due to the difficulty of intravenous line establishment for infants. However, no sepsis studies focusing on young infants have been conducted to date, and it is essential to explore these elements for successful sepsis management that is specific to febrile young infants. 

Therefore, it is critical to examine the factors that contribute to the completion of the sepsis bundle in infants with suspected septic shock to improve the initial therapy quality and mortality of septic infants. This study aims to identify the predictor of the successful completion of a 1-h sepsis bundle. 

## 2. Results

### 2.1. Demographics

Among 323 patients registered in the febrile young infant database, 118 who were suspected of septic shock and who received antibiotics were analyzed for this study (Figure 1). The median age of the patients was 37 days (interquartile range: IQR 21.0–65.0), and 63 (53.4%) were male. Thirty-eight patients (32.2%) received a bundle-compliant treatment. Their median core body temperature, heart rate, respiratory rate, and pulse oximetry were 38.6 °C (IQR 0.5), 178 beats per minute (IQR 164.0–192.2), 38.5 breaths per minute (IQR 31.0–47.3), and 99.0% (IQR 97–100), respectively. The median white blood cell count (×10^3^), C-reactive protein (CRP), and lactate level were 9.0 (IQR 6.5–12.6), 0.5 mg/dL (IQR 0.2–1.3), and 2.6 mmol/L (IQR 1.9–3.3), respectively. The proportion of final diagnoses of serious bacterial infections and serious viral infections was 11.9% (*n* = 14) and 24.6% (*n* = 29), respectively. The median hospital stay was 6.0 days (IQR SD 5.0–7.0), and eight patients (6.8%) were admitted to the intensive care unit (Table 1).

### 2.2. Comparison between the Patient Group with Bundle Completion and without Bundle Completion

In the univariate analysis, patients who received bundle-compliant (BC) care were compared with those who received non-bundle-compliant (non-BC) care; demographic data, such as body weight (4.8 kg vs. 4.2 kg, *p* = 0.48), age (40.0 days vs. 31.0 days, *p* = 0.24), and sex (male ratio 65.8% vs. 48.8%, *p* = 0.11), were similar.

Regarding the patients’ vital signs, the BC patients presented a higher core body temperature (38.9 °C (IQR 38.4–39.0) vs. 38.5 °C (IQR 38.2–38.9), *p* = 0.05) and tachycardia (187.5 beats per minute (IQR 169.0–198.5) vs. 173.0 beats per minute (IQR 158.0–188.8), *p* < 0.01). The respiratory rate (39.0 breaths per minute (IQR 29.8–51.0) vs. 38.5 breaths per minute (IQR 32.0–45.8), *p* = 0.72) and oxygen saturation (99.0% (IQR 97.8–100) vs. 99.0% (IQR 97.0–100), *p* = 0.26) were similar between the two groups. While the proportion of the patients with prolonged capillary refilling time (CRT) was significantly higher in the BC group (73.7% vs. 38.8%, *p* < 0.01), there were no differences in the proportion of the patients with cool extremities, hypotension, and Glasgow Coma Scale (GCS) less than 15 between the two groups. Although there was no difference in the CRP and venous lactate value between the two groups, the WBC was significantly lower in the BC patients than it was in the non-BC patients (6.9 (IQR 5.5–10.2) vs. 9.9 (IQR 7.5–13.6), *p* < 0.0). The prevalence of SBIs (13.2% vs. 11.3%, *p* = 0.77) and SVIs (31.6% vs. 21.3%, *p* = 0.26) was similar between the two groups.

The proportion of patients who were admitted to the intensive care unit (13.2% vs. 3.8%, *p* = 0.11) was not statistically different between the two groups. While the patients in the BC group received a larger volume of infusion fluid (49.5 mL/kg vs. 11.2 mL/kg, *p* < 0.01), the length of the hospital stay (5.5 days (IQR 5.0–8.3) vs. 6.0 days (IQR 5.0–6.0) was similar between the two groups (Table 2).

### 2.3. Predictor (s) of the 1 h Bundle Completion

Among these variables, we selected the variables age, male sex, heart rate, and respiratory rate for the logistic regression analysis. In the multiple logistic regression, heart rate (odds ratio: OR 1.02, 95% CI: 1.00–1.04) was identified as a significant predictor of the 1-h sepsis bundle (Table 3). This indicates that a 1-beat increase in the heart rate increases the likelihood of sepsis bundle completion by a 1.02-fold rise and that a 10-beat increase in heart rate increases the likelihood by a 1.23-fold rise (1.02 to the 10th power).

The odds ratios of male sex (OR 2.00, 95%CI: 0.87–4.60), age (OR 1.00, 95%CI 0.99–1.02), and respiratory rate (OR 1.00, 95%CI, 0.96–1.03) were not statistically significant (Table 3).

## 3. Discussion

This study aimed to explore the predictor of the successful completion of a 1-h sepsis bundle for young infants with suspected septic shock and found that elevated heart rate is a significant positive predictor of adherence to the bundle. Namely, the higher the patient’s heart rate, the more sepsis recognition is enhanced; for example, a 10-beat elevation in heart rate increases the likelihood of sepsis bundle completion by 1.23-fold, and a 20-beat elevation in heart rate increases the likelihood by 1.52-fold.

Studies have suggested that compliance with the 1-h sepsis bundle is associated with favorable outcomes in children with septic shock [9,10,11,12]. However, no sepsis studies on young infants have been performed even though infants are the most susceptible to sepsis in pediatric emergency and critical care. The current study shows, for the first time, the “surrogate” indicator of the favorable outcomes in treating infants with sepsis in emergency department (ED) and further proposes that compliance with the 1 h bundle can be improved by increasing the awareness of tachycardia in febrile infants. The heart rate of young infants is often overlooked in emergency care, as this vital sign parameter is easily affected by the patients’ status, such as crying and high-grade fever. Nevertheless, our findings emphasize the importance of continuous cardiac monitoring to achieve the early recognition and intervention for septic young infants in EDs. 

While it is more difficult to establish an intravenous line in younger infants than it is in older children, age was not a significant negative predictor of sepsis bundle completion in the analysis. The completion of the bundle is the outcome of the team’s performance. Securing the intravenous line is just one skill component of sepsis resuscitation, and the collaborative performance of each team member and shared decision-making are more important to achieve the 1-h sepsis bundle. In this vein, tachycardia served as the “alert” to the resuscitation team, promoting the shared responsibility of the members and ultimately facilitating comprehensive team performance. Balamuth suggested that the vital sign-based electronic alert effectively improved the recognition of septic shock in children [13]. Building on this finding, our study implies that vital signs, such as heart rate, enable the resuscitation team to recognize sepsis promptly and to improve the team’s performance. Heart rate is one of the most sensitive vital sign parameters for treatment intervention during resuscitation based on the sepsis bundle; therefore, the improvement of tachycardia can be a goal that is shared by all team members. In this context, heart rate works as a helpful indicator that is consistently used in the sepsis recognition and resuscitation phases for patients. 

There are several limitations to this study. First, we only performed this study at a single tertiary care institution in Japan. In addition, the sample size of this study was small; thus, there is a possibility of a type II error in the univariate statistical analysis. However, the smaller size of this report was due to the study design focusing on infants, which is the unique aspect of our findings. Multicentered studies are needed to verify the generalizability of our findings. 

## 4. Materials and Methods

### 4.1. Study Design and Setting

This is a secondary analysis study using a prospective cohort registry collected at Tokyo Metropolitan Children’s Medical Center, a pediatric tertiary care center in Tokyo, Japan [14]. The data registration duration was from 1 August 2014 to 30 September 2016. There were approximately 38,000 annual pediatric visits to the ED of the hospital. While the initial patient management is usually provided by pediatric or emergency medicine residents and pediatric emergency medicine fellows, board-certificated emergency medicine physicians or pediatricians supervise patient care [15,16]. Patients with sepsis or suspected sepsis were treated based on the institutional protocol by following the International Pediatric Sepsis Consensus Conference (2005) [17] and Surviving Sepsis Campaign: International Guidelines for Management of Sepsis and Septic Shock (2012) [18].

### 4.2. Participants

Our febrile young infant database included all young infants (aged < 90 days) who had visited our ED with a core body temperature of 38.0 °C or higher and 36.0 °C or lower at triage. To select the suspected septic shock patients, we excluded those who did not meet the definitions of septic shock or suspected septic shock as per the updated guidelines and recent study [8,19] and those infants who did not receive antibiotics in the ED. Septic shock was defined as severe infections showing cardiovascular dysfunction, including hypotension (systolic blood pressure < 70 mmHg) and impaired perfusion [8,9,10], and suspected septic shock was identified if there was a delayed capillary refill time (>2 s) or cool extremities as well as an altered mental status (GCS < 15) [19]. 

### 4.3. Data Collection

We prospectively collected information on patient demographics, including patient age, sex, vital signs (i.e., core body temperature, heart rate, respiratory rate, oxygen saturation, physical examination findings, and GCS), laboratory examination results (i.e., WBCs, CRP, and venous lactate value), definitive diagnosis (SBIs or SVIs), a specialty of first-touch physicians, compliance to the 1-h sepsis bundle, volume of fluid resuscitation, and length of hospital stay. To measure the vital signs, the core body temperature was measured at triage before the physician examinations. We adopted the minimum values for the respiratory and heart rate within the first 30 min of initial physician care to eliminate the influence of patient agitation or crying.

### 4.4. Statistical Analysis

Data were analyzed using the SPSS software package, version 23 (Armonk, NY, USA: IBM Corp.). Standard descriptive statistics were reported with the mean and SD for the continuous variables and with the frequency and percentage for the categorical variables. A Chi-squared test, Fisher’s exact test, and the Mann–Whitney U test were conducted for statistical analysis, and a two-sided *p* < 0.05 value was considered to be statistically significant. We also conducted a logistic regression analysis to identify the predictor of the achievement of antibiotic administration within 1 h after the recognition of sepsis. The independent variables of the regression model were selected based on the literature review and the interests of the researchers.

### 4.5. Ethics

The ethical committee of Tokyo Metropolitan Children’s Medical Center approved this study (approval number: H26-15). Patient consent was obtained on an opt-out basis.

## 5. Conclusions

We identified that tachycardia is a predictor for the successful completion of a 1-h sepsis bundle for young infants with suspected septic shock and is a possible indicator of improving team-based sepsis management quality.

## Figures and Tables

**Figure 1 antibiotics-10-01414-f001:**
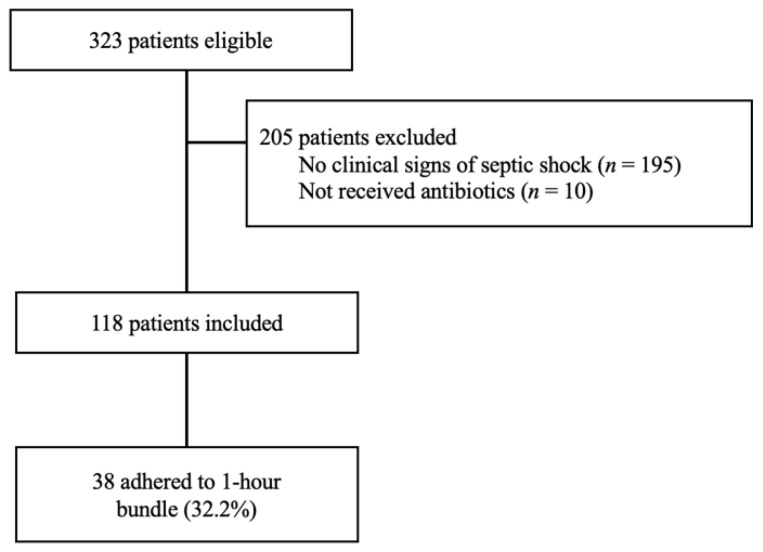
Study flowchart.

**Table 1 antibiotics-10-01414-t001:** Clinical characteristics of patients (*n* = 118).

Variables	
Demographics	
Age, median (IQR), Days	37.0 (21–65)
Wt, median (IQR), kg	4.3 (3.6–5.3)
Male, % (*n*)	53.4 (63)
Vital signs	
Core body temperature, median (IQR), °C	38.6 (38.3–38.9)
Heart rate, median (IQR), beats per minute	178.0 (164.0–192.2)
Respiratory rate, median (IQR), breaths per minute	38.5 (31.0–47.3)
Pulse oximetry, median (IQR), %	99 (97–100)
Glasgow Coma Scale lower than 15, % (*n*)	16.1 (19)
Physical examinations	
Cool extremities, % (*n*)	71.2 (84)
Prolonged Capillary Refilling Time, % (*n*)	50.0 (59)
Hypotension, % (*n*)	8.5 (10)
Laboratory examinations	
WBC count, median (IQR), ×10^3^	9.0 (6.5–12.6)
Neutrophils, median (IQR), %	48.8 (34.1–62.1)
Lactate, median (IQR), mmol/L	2.6 (1.9–3.3)
CRP, median (IQR), mmol/L	0.5 (0.2–1.3)
Platelet count, median (IQR), ×10^3^	38.6 (28.6–48.6)
Bilirubin, median (IQR), mg/dl	3.5 (1.8–10.1)
Creatinine, median (IQR), mg/dl	0.24 (0.21–0.27)
Clinical diagnosis	
Serious bacterial infections, % (*n*)	11.9 (14)
Serious viral infections, % (*n*)	24.6 (29)
Pediatric SOFA score, median (IQR)	2 (1–3)
Treatment	
Hospitalization inwards, % (*n*)	93.2 (110)
Intensive care unit admission, % (*n*)	6.8 (8)
Day of stay, median (IQR), days	6.0 (5.0–7.0)
Bundle completions, % (*n*)	32.2 (38)
Total-infusion amount, median (IQR), mL/kg	19.8 (9.8–37.6)

Note. WBC, white blood cell; CRP, C-reactive protein; pSOFA, pediatric sequential organ failure assessment; IQR, interquartile range.

**Table 2 antibiotics-10-01414-t002:** Comparison between patients with bundle completion and the incompletion group.

Variables	Bundle Completion (*n* = 38)	Bundle Incompletion (*n* = 80)	*p*-Value
Demographics			
Bodyweight, median (IQR), kg	4.8 (3.7–5.3)	4.2 (3.5–5.3)	0.48
Age, median (IQR), days	40.0 (28.3–60.3)	31.0 (20.0–65.8)	0.24
Male, % (*n*)	65.8 (25)	48.8 (39)	0.11
Vital signs			
Core body temperature, median (IQR), °C	38.9 (38.4–39.0)	38.5 (38.2–38.9)	0.05
Heart rate, median (IQR), beats per minute	187.5 (169.0–198.5)	173.0 (158.0–188.8)	<0.01
Respiratory rate, median (IQR), breaths per minute	39.0 (29.8–51.0)	38.5 (32.0–45.8)	0.72
Pulse oximetry, median (IQR), %	99.0 (97.8–100)	99.0 (97.0–100)	0.26
Physical examinations			
Cool extremities, % (*n*)	31 (81.6)	53 (66.3)	0.13
Prolonged Capillary Refilling Time, % (*n*)	28 (73.7)	31 (38.8)	<0.01
Hypotension, % (*n*)	3 (7.9)	7 (8.8)	1.00
Glasgow Coma Scale lower than 15, % (*n*)	23.7 (9)	12.5 (10)	0.18
Laboratory examinations			
WBC count, median (IQR), ×10^3^	6.9 (5.5–10.2)	9.9 (7.5–13.6)	<0.01
Neutrophils, median (IQR), %	46.1(30.6–59.2)	50.4 (35.5–62.4)	0.29
Lactate, median (IQR), mmol/L	2.4 (1.9–3.1)	2.7 (1.9–3.4)	0.36
CRP, median (IQR), mg/dl	0.5 (0.2–1.1)	0.5 (0.2–1.6)	0.86
Platelet count, median (IQR), ×10^3^	35.8 (27.5–47.8)	39.5 (30.2–51.0)	0.47
Bilirubin, median (IQR), mg/dl	2.8 (1.5–7.2)	3.8 (1.1–10.7)	0.92
Creatinine, median (IQR), mg/dl	0.24 (0.20–0.26)	0.24 (0.21–0.27)	0.57
Clinical diagnosis			
pSOFA score, median (IQR)	2 (1–3)	2 (0–3)	0.71
Serious bacterial infections, % (*n*)	13.2 (5)	11.3 (9)	0.77
Serious viral infections, % (*n*)	31.6 (12)	21.3 (17)	0.26
Treatment			
Intensive care unit admission, % (*n*)	13.2 (5)	3.8 (2)	0.11
Total-infusion amount, median (IQR), mL/kg	49.5 (33.2–57.0)	11.2 (7.6–20.2)	<0.01
Length of stay, median (IQR), days	5.5 (5.0–8.3)	6.0 (5.0–6.0)	0.61

Note. WBC, white blood cell; CRP, C-reactive protein; pSOFA, pediatric sequential organ failure assessment, IQR, interquartile range.

**Table 3 antibiotics-10-01414-t003:** Predictors of 1 h bundle completion.

Characteristics	Odds Ratio	95% CI	*p*
Heart rate	1.02	1.00–1.04	0.04
Male sex	2.00	0.87–4.60	0.11
Age	1.00	0.99–1.02	0.68
Respiratory rate	1.00	0.96–1.03	0.88

## Data Availability

The data that support the findings of this study are available from the corresponding author upon reasonable request.

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
