# Peer review of "Predictor of Early Administration of Antibiotics and a Volume Resuscitation for Young Infants with Septic Shock"

_antibiotics, 2021, doi:10.3390/antibiotics10111414_

Round 1
Reviewer 1 Report
The authors describe within their secondary analysis of a prospective cohort study predictors for the use of 1-h bundle in young infants with septic shock. Even the theme of the study seems to be of high interest, there are several points within the study design and performance limiting the results and conclusions.
Methods:
The selection criteria of the patients with suspected septic shock are insufficient. The authors used only clinical signs. At least white blood cell counts (WBC) needs to be taken in account for selecting the patients out of the database (compare e.g. Clin Perinatol. 2010 Jun; 37(2): 439–479.) . Moreover, the definitions for infants with septic shock include also a lower body temperature then 36°C, which has not been recognised by the authors. Therefore, a reanalyses of the hole database in relation the whole definitions of septic shock in infants is necessary.
Within the statistical analysis, no test was performed to prove the normal distribution of the analysed cohort. This in terms is necessary to use ANOVA analyses as well as mean value with standard deviation.
Results:
The presented results show nearly normal white blood cell counts (WBC) a less proven bacterial and viral infections underlining the above described methodical faults.
Moreover, the low rates of treatment by an emergency physician are less explainable. Does it mean, that at least 35 – 50% of the included infants have not be treated by a physician? This is quite irritating.
Regarding the results section, presentation of inflammation markers (e.g. CRP, IL-8) would also improve the quality and comprehensibility for the reader.
Author Response
Thank you very much for your insightful review.
Comment1
“The selection criteria of the patients with suspected septic shock are insufficient. The authors used only clinical signs. At least white blood cell counts (WBC) needs to be taken in account for selecting the patients out of the database (compare e.g., Clin Perinatol. 2010 Jun; 37(2): 439–479.).
Response
Thank you for your critical suggestions. As the reviewer pointed out, white blood cell counts has been utilized to define septic shock in the field of neonatology and pediatric emergency medicine. However, the recent guideline (Weiss et al., PCCM 2020) defines “septic shock in children as severe infection leading to cardiovascular dysfunction (including hypotension, need for treatment with a vasoactive medication, or impaired perfusion)." Thus, the updated guideline focuses on the organ damages indicated by physical and laboratory findings rather than inflammatory markers such as WBCs and CRP, and this approach is commonly utilized in recent papers on pediatric sepsis (Lane et al., Hosp. Pediatr. 2020.). In response to your suggestions, we clarify the definitions of sepsis (Line 180-186). We would appreciate your understanding.
Comment2
Moreover, the definitions for infants with septic shock include also a lower body temperature then 36°C, which has not been recognised by the authors. Therefore, a reanalyses of the hole database in relation the whole definitions of septic shock in infants is necessary.”
Response
I agree with you regarding hypothermic sepsis. We added two patients with hypothermia (30.0 °C and 35.0 °C) in the database and conducted re-analysis (Line 22 and 179).
Comment 3
“Within the statistical analysis, no test was performed to prove the normal distribution of the analysed cohort. This in terms is necessary to use ANOVA analyses as well as mean value with standard deviation. The presented results show nearly normal white blood cell counts (WBC) a less proven bacterial and viral infections underlining the above described methodical faults.”
Response
Thank you for your comment. We added two hypothermia patients in response to your comment above (comment 2), and some of the data of the patients (i.e., body temperature and heart rate) are outlier variables. As a result, we consider that the assumption of normal distribution in continuous variables of this data has been violated. To deal with this issue, we changed the statistical test for the continuous variables to the Mann-Whitney U test for aiming for more robust analysis and re-wrote all the results (Line 203 and all of the result section). Thank you so much for your suggestion.
Comment 4
“The low rates of treatment by an emergency physician are less explainable. Does it mean, that at least 35 – 50% of the included infants have not be treated by a physician? This is quite irritating.”
Response
Thank you for pointing this out. This means that the emergency physicians treated 55% of the patients, and pediatricians treated the other patients (45%). My apology for the confusing descriptions. Since this data does not add significant information to this study's results, we removed this information from the table to avoid readers' confusion.
Comment 5
“Regarding the results section, presentation of inflammation markers (e.g. CRP, IL-8) would also improve the quality and comprehensibility for the reader.”
Response
Thank you for your suggestion. We added CRP data in the tables and manuscript (Table 1&2, Line 77-78 and 100).
Reviewer 2 Report
This paper is current and interesting; however, I have a few comments and suggestions for the authors.
Materials and Methods section should follow the Introduction section, before Results section. Also, Figure 1. Study flowchart should be removed from Results section and included in subsection Participants.
There are some paragraphs that should be re-checked/rephrased:
“We found that tachycardia facilitated the sepsis recognition and promoted of the successful completion of a 1-h sepsis bundle for young infants with suspected septic shock and an essential indicator improving the quality of the sepsis resuscitation team’s performance.”
“Some research groups have aimed to develop prediction models including clinical variables such as age, while blood cell count (WBC), and urine test results for detecting serious bacterial infections (SBIs).”
“It is also important to create evidence to improve emergency care for critically ill infants with sepsis.” – my suggestion is to replace create with find or uncover
“This study aimed to explore the predictor of the successful completion of a 1-h sepsis bundle for young infants with suspected septic shock and found that elevated heart rate is a significant positive predictor of the adherence of the bundle.” – of should be replaced with to
“We prospectively collected information on patient demographics, including patient age, sex, vital signs (i.e., core body temperature, heart rate respiratory rate, oxygen saturation, and Glasgow Coma Scale: GCS), laboratory examination results (i.e., WBCs and venous lactate value), definitive diagnosis (SBIs or SVIs), a specialty of first-touch physicians, compliance to the 1-h sepsis bundle, volume of fluid resuscitation, and length of hospital stay.” – a comma should be inserted between heart rate and respiratory rate
Author Response
Thank you very much for your careful review.
Comment1
“Materials and Methods section should follow the Introduction section, before Results section. Also, Figure 1. Study flowchart should be removed from Results section and included in subsection Participants.”
Response
Thank you for your suggestion. We follow the instruction of this journal (https://www.mdpi.com/journal/antibiotics/instructions) and would like to leave this point to the editor to decide.
Comment 2
“We found that tachycardia facilitated the sepsis recognition and promoted of the successful completion of a 1-h sepsis bundle for young infants with suspected septic shock and an essential indicator improving the quality of the sepsis resuscitation team’s performance.”
“Some research groups have aimed to develop prediction models including clinical variables such as age, while blood cell count (WBC), and urine test results for detecting serious bacterial infections (SBIs).”
“It is also important to create evidence to improve emergency care for critically ill infants with sepsis.” – my suggestion is to replace create with find or uncover
“This study aimed to explore the predictor of the successful completion of a 1-h sepsis bundle for young infants with suspected septic shock and found that elevated heart rate is a significant positive predictor of the adherence of the bundle.” – of should be replaced with to
“We prospectively ……(i.e., core body temperature, heart rate respiratory rate, oxygen saturation, and Glasgow Coma Scale: GCS),…..” – a comma should be inserted between heart rate and respiratory rate
Response
Thank you very much for your careful observation. We corrected the points you kindly suggested.
Reviewer 3 Report
I agree that early administration of antibiotics is critical in sepsis treatment for young children.
However some comments have to be made:
Figure 1: I cannot understand how the numbers come about. (321 – 137 = 116 ??) Furthermore the numbers in the text differ (fourty-eight?) 115 ? Please check that!
Table 2 and 3: Please add SOFA score to the table! Add all signs to the table, which were utilized for the diagnosis of septic shock. (cardiovascular dysfunction, hypotension, impaired perfusion, delayed capillary refill time, mental status and temperature of extremities.
When were data like core body temperature and heart rate, respiratory rate eg. measured?
Table 3 has to be renewed after addition of the missing parameters.
Author Response
Comment 1
“Figure 1: I cannot understand how the numbers come about. (321 – 137 = 116 ??) Furthermore, the numbers in the text differ (forty-eight?) 115? Please check that!”
Response
Thank you for pointing this out. We corrected the figures and the text (Figure and Line 24 and 70-71).
Comment 2
“Table 2 and 3: Please add SOFA score to the table! Add all signs to the table, which were utilized for the diagnosis of septic shock. (cardiovascular dysfunction, hypotension, impaired perfusion, delayed capillary refill time, mental status and temperature of extremities.”
Response
Thank you for your suggestion. We added these variables in the tables (Table 1 and 2).
Comment 3
“When were data like core body temperature and heart rate, respiratory rate eg. measured?”
Response
Thank you for your suggestion. We described the timing and method of the measurement for vital signs (Line 193-197).
Round 2
Reviewer 1 Report
The authors present a revised version of their manuscript. Within this version significant changes and improvement of the presented data have been performed, so that most of the points, judged within the first review round have been addressed. Nevertheless, the authors present within this version a manuscript with a relevant number of spelling errors, which needs to be corrected ( e.g. Table 1 Variavles instead of Variables, Capirally instead of Capilarry, etc.).
Author Response
Thank you very much for your observation.
We corrected all of the spelling errors.
Reviewer 3 Report
The authors improved the manuscript in the revision. However lots of spelling errors can be found, wich have to be extensively studied.
Author Response
Thank you very much for your careful observation.
We corrected all of the spelling errors.